# Capturing ion trapping and detrapping dynamics in electrochromic thin films

Renfu Zhang [ORCID][1], Qinqi Zhou[1], Siyuan Huang [ORCID][1], Yiwen Zhang [ORCID][1] & Rui-Tao Wen [ORCID][1,2] ✉

Ion trapping has been found to be responsible for the performance degradation in electrochromic oxide thin films, and a detrapping procedure was proved to be effective to rejuvenate the degraded films. Despite of the studies on ion trapping and detrapping, its dynamics remain largely unknown. Moreover, coloration mechanisms of electrochromic oxides are also far from clear, limiting the development of superior devices. Here, we visualize ion trapping and detrapping dynamics in a model electrochromic material, amorphous $WO_3$. Specifically, formation of orthorhombic $Li_2WO_4$ during long-term cycling accounts for the origin of *shallow* traps. *Deep* traps are multiple-step-determined, composed of mixed $W^{4+}$-$Li_2WO_4$, amorphous $Li_2WO_4$ and $W^{4+}$-$Li_2O$. The non-decomposable $W^{4+}$-$Li_2WO_4$ couple is the origin of the *irreversible* traps. Furthermore, we demonstrate that, besides the typical small polaron hopping between $W^{5+} \leftrightarrow W^{6+}$ sites, bipolaron hopping between $W^{4+} \leftrightarrow W^{6+}$ sites gives rise to optical absorption in the short-wavelength region. Overall, we provide a general picture of electrochromism based on polaron hopping. Ion trapping and detrapping were demonstrated to also prevail in other cathodic electrochromic oxides. This work not only provides the ion trapping and detrapping dynamics of $WO_3$, but also open avenues to study other cathodic electrochromic oxides and develop superior electrochromic devices with great durability.

Electrochromic materials are able to change their optical properties under the action of an electric field[1,2]. This unique property gives electrochromic devices a wide range of applications in information displays, adjustable mirrors, variable emittance surfaces and smart windows[3–5], and the latter is especially attractive in decarbonlization due to the combined advantages of energy-saving in buildings while maintaining indoor comfort[6–10]. Amorphous tungsten oxide (*a*-$WO_3$) is the most studied electrochromic material, and almost all oxide-based electrochromic devices employ $WO_3$[1,11–14]. As many other electrochromic oxides possess similar coloration dynamics, $WO_3$ represents a widely used model for exploring the electrochromic processes[1,2,15–17]. Small *polaron* hopping is the most accepted theory[1,15,18–20], which

asserts that as an ion and an electron are simultaneously inserted, the electron is localized around a $W^{6+}$ site and forms a $W^{5+}$ one. The localized electron is named a *polaron* and optical absorption then arises from polaron hopping between the $W^{5+}$ site and a nearby $W^{6+}$ site through[15,21]:

$$W_{(a)}{}^{5+} + W_{(b)}{}^{6+} \overset{h\nu_1}{\leftrightarrow} W_{(a)}{}^{6+} + W_{(a)}{}^{5+} \qquad (1)$$

Despite the fact that the phenomenon of electrochromism has been discovered for over 50 years[22–24], and associated research as well as commercialization also keep growing with enhanced durability of $WO_3$[25–27], performance degradation upon cycling is inevitable and its

[1]Department of Materials Science and Engineering, Southern University of Science and Technology, Shenzhen 518055, China. [2]Guangdong Provincial Key Laboratory of Functional Oxide Materials and Devices, Southern University of Science and Technology, Shenzhen 518055, China. ✉e-mail: Wenrt@sustech.edu.cn

origin is still far from clear[24,28–32], which largely limits the development of superior devices. The ion trapping/detrapping phenomenon reported a few years ago provides a new vista to reframe the electrochromism field and also pave the way to extend the lifetime of $WO_3$ thin films[28,33–35]. It has been found that failure to extract the inserted ions in a single cycle leads to ion accumulation in the electrodes as cycling proceeds[16,17,28,30,35,36]. The ion accumulation is termed "ion trapping" and erodes electrochromic performance. Specifically, three different types of traps are found to degrade the electrochromic performance[28,33]: (i) *shallow traps* which degrade the colored state only; (ii) *deep traps* that degrade both colored and bleached states; and (iii) *irreversible traps*, in which the resided ions cannot be released any longer, degrade the short-wavelength region of optical transmittance and are considered to be a minor effect. Although extensive studies have been made[33,37–42] and the ion detrapping procedure is shown to be effective to regain initial electrochromic performance, the origin of the *traps* and associated dynamics remain unknown which largely limits the understanding of *traps* and further attempts for trap suppression or even elimination.

Here we show direct experimental evidence of origins of all three kinds of traps that are present during ion trapping and detrapping processes. S*hallow* traps are due to a phase change from amorphous tungsten bronze ($Li_xWO_3$) to orthorhombic lithium tungstate ($Li_2WO_4$) which largely suppresses the hopping from $W^{5+} \leftrightarrow W^{6+}$ sites. Compare to $WO_3$, the extra oxygen in an orthorhombic $Li_2WO_4$ molecular is from the electrolyte. Details of dynamics for deep traps have also been unveiled, specifically, a rejuvenation from *deep* traps is a multiple-step-determined process and the ion-releasing sequence starts from decomposition of coupled $W^{4+}$-$Li_2O$, amorphous $Li_2WO_4$ and coupled $W^{4+}$-$Li_2WO_4$, revealing a complex structure reconfiguration in the host matrix. *Irreversible* traps are coupled $W^{4+}$-$Li_2WO_4$ and are non-decomposable. In addition to the small *polaron* hoping between $W^{5+}$ and $W^{6+}$ sites, optical absorption in the short-wavelength region is due to *bipolaron* hopping between $W^{4+}$ and $W^{6+}$ sites, after formation of $W^{4+}$ upon reversible intercalation or ion trapping process. Three forms of $W^{4+}$ sites, reversible and irreversible, are also identified. Overall, our work unveils the dynamics of ion trapping and detrapping, and provides a general picture of electrochromism based on *polaron* hopping, in the representative electrochromic oxide, $WO_3$. Electrochromism of the cathodic oxides (such as $WO_3$, $MoO_3$, $Nb_2O_5$, $Ta_2O_5$ and $TiO_2$) is based on small ion intercalation and considered to obey polaron hopping[1,15,43–46]. Meanwhile, ion-trapping induced degradation is a ubiquitous phenomenon in cathodic electrochromic oxides[16,17,28,46]. For example, in addition to the model electrochromic material, $WO_3$[28,33], ion trapping and detrapping also prevail in the rest of cathodic electrochromic oxides, i.e., $MoO_3$, $TiO_2$, $Nb_2O_5$ and $Ta_2O_5$. Therefore, our findings here not only pave the way to develop superior electrochromic devices based on $WO_3$, but also serves as the guideline to explore the degradation dynamics for other cathodic electrochromic oxides. The methodologies of a combination of spectroelectrochemistry and other spectro- and microscopic characterizations can be well expanded to other electrochromic materials and ion intercalated systems.

## Results and discussion

### Performance degradation and rejuvenation

We start from the most used potential window for $WO_3$, i.e., 2.0–4.0 V vs Li[25–27]. Figure 1a shows cyclic voltammetry (CV) curves of $WO_3$, indicating a typical amorphous structure. The envelope area of the CV curves, from which inserted and extracted charge in each cycle can be calculated, are reducing as cycling proceeds. The inserted charge is larger than the extracted one, and this leads to ion accumulation in the electrode and decreases the reversibility (the difference between inserted and extracted charge in each cycle is shown in Supplementary Fig. 1). Open circuit potential (OCP) drops from 3.3 V at the pristine

state to 3.1 V after 1000 cycles and the associated electrochromic performance (i.e., optical modulation, $\Delta T$, defined as the transmittance in the bleached state ($T_b$) minus the one in the colored state ($T_c$).) exhibits a clear degradation (Fig. 1c). It can also be noted from the in-situ optical transmittance that degradation of $\Delta T$ only arises from a degraded colored state, whereas the bleached state maintains the same transmittance as the pristine state (Fig. 1c for the wavelength 550 nm and Fig. 1g for the full spectrum from 380 nm to 2250 nm; transmittances at other wavelengths can be seen in Supplementary Fig. 2). The coloration is due to typical *polaron* hopping between $W^{5+}$ and $W^{6+}$ sites when electrons are injected as described in Eq. 1 and the optical transmittance is basically flat except for a broad peak at the short-wavelength region. Bear in mind that the origin of the degradation of the colored state is as ascribed to *shallow* traps[28,33] which evenly lift-up the full spectrum at the colored state upon cycling. This indicates that resided ions in *shallow* traps do not contribute to optical absorption[28,32,33] but only "*suppress*" the $W^{6+} \leftrightarrow W^{5+}$ hopping.

A so-called galvanostatic "detrapping" procedure was conducted after 1000 cycles, namely, a constant current of 3.0 μA cm$^{-2}$ was applied to the degraded film aiming to regain the initial electrochromic performance (schematic diagram of detrapping, and other constant currents used for detrapping were also compared, as shown in Supplementary Fig. 3). Optical transmittance in the bleached state remains constant throughout the entire "detrapping" process. Performing subsequent 10 CV cycles after "detrapping" clearly demonstrates that both charge capacity and optical modulation are completely rejuvenated (Fig. 1c and g). Moreover, the detrapping procedure from *shallow* traps is a continuous process[33] and is qualitatively different from the situation for *deep* traps, as will be shown later.

The rejuvenated film was then subjected to cycling in the range 1.5–4.0 V (Fig. 1b) with a purpose of studying on the practicability to extend the potential window of $WO_3$ and the case of significant ion trapping. Charge capacity (Fig. 1e and Supplementary Fig. 1b) as well as optical transmittance (Fig. 1c and enlarged view in Fig. 1f) degrade more rapidly than for the case between 2.0 and 4.0 V, indicating severe ion trapping in the electrode and eroded electrochromic activity. The OCP drops from 3.3 V to 2.4 V after 20 cycles, and more importantly, optical transmittances at both colored and bleached states degrade. The origin of degradation of both colored and bleached states is ascribed to *deep* traps. Clearly, ions residing in *deep* traps are responding in a different manner as compared to *shallow* traps. Moreover, the broad transmittance peak at the short-wavelength region becomes flat (Fig. 1h, 1st colored, switching process can be seen from Supplementary Movie 1) or even shows a valley (Supplementary Fig. 4) while the long-wavelength region increases. This shows that the *polaron* hopping mode in the electrode has also changed in the range 1.5–4.0 V.

An analogous "detrapping" procedure was conducted to release the ions from *deep* traps. Three distinct steps can be noted from optical transmittance profiles (green area in Fig. 1c): step one, full optical spectrum slightly lifts up at the very beginning of the detrapping process; step two, full spectrum maintains as detrapping proceeds; step three, a sharp increase of the transmittance reaching to its pristine state, except in the short-wavelength region. The variation of transmittance in the whole spectral range (Supplementary Fig. 5) is consistent with the single-wavelength (550 nm) data except for the unrecovered part in the short wavelength region (350 nm–550 nm), whose origin is ascribed to *irreversible* traps. Here, we confirm that ions residing in *deep* traps can also be released whereas ions in *irreversible* traps become permanently immobile. Moreover, the potential profiles during galvanostatic detrapping processes (Fig. 1i), also indicate that ions released from *shallow* and *deep* traps are distinctly different. Consistent results can also be found when detrapping is carried out through a potentiostatic process (Fig. 1j), where the current density shows a peak, analogous to the apparent valley of the potential profile

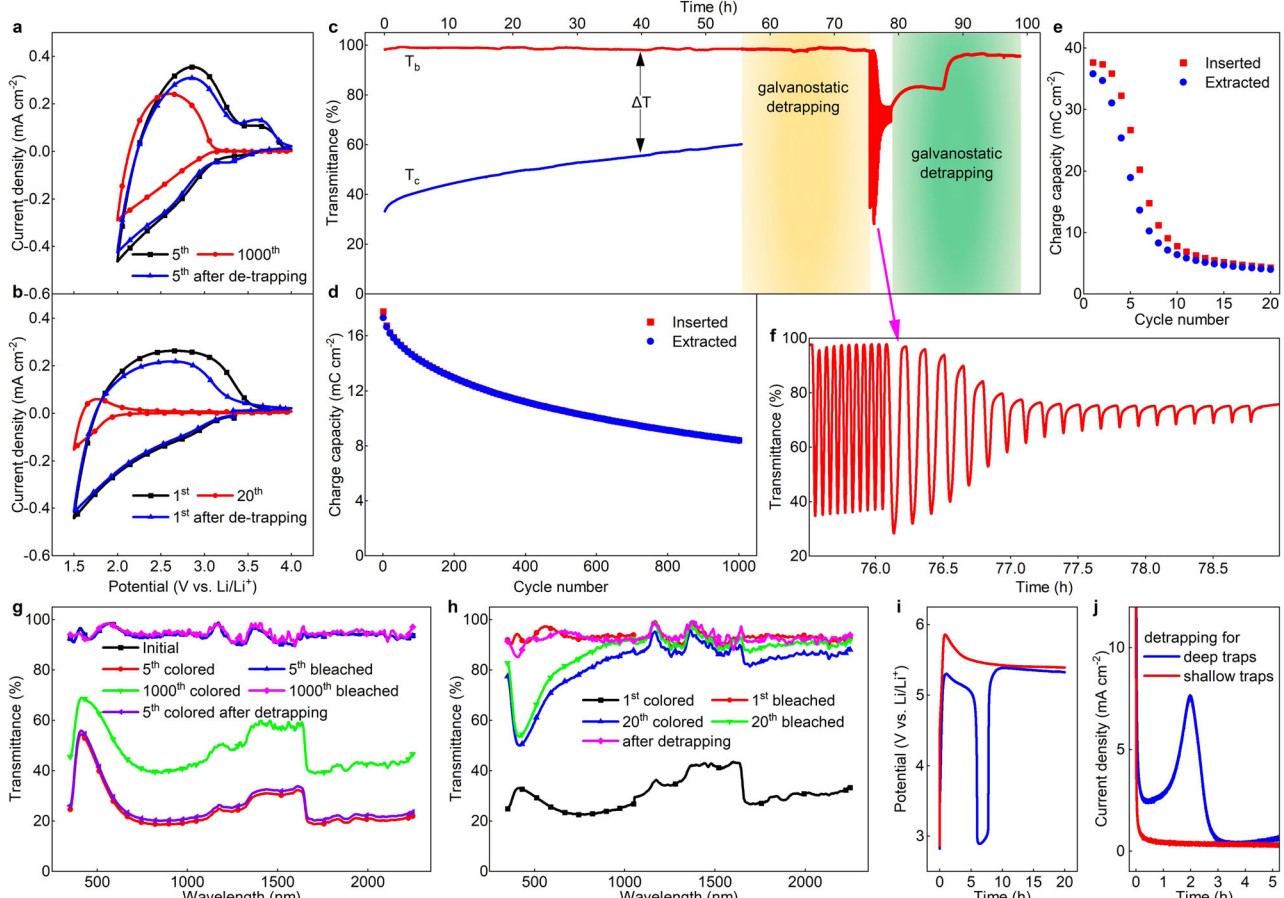

**Fig. 1 | Spectroelectrochemistry of WO₃ during ion trapping and detrapping process. a, b** CV data for various cycle numbers at 2.0–4.0 V (20 mV s⁻¹) and 1.5–4.0 V (10 mV s⁻¹), respectively. **c** In-situ optical transmittance at 550 nm upon various ion exchange processes. The sequence is 1000 CVs at 2.0–4.0 V (20 mV s⁻¹), galvanostatic detrapping for 20 h, 10 CVs at 2.0–4.0 V (20 mV s⁻¹), 20 CVs at 1.5–4.0 V (10 mV s⁻¹) and galvanostatic detrapping for 20 h. Optical modulation, $\Delta T$, defined as the transmittance in the bleached state ($T_b$) minus the one in the colored state ($T_c$). **d, e** Charge capacity variation at 2.0–4.0 V (20 mV s⁻¹), and 1.5–4.0 V

(10 mV s⁻¹), respectively. **f** Enlarged view of Fig. 1c for 10 CVs in the range of 2.0–4.0 V (20 mV s⁻¹) after galvanostatic degrapping and rapid degradation in the following 20 CVs in the range of 1.5–4.0 V (10 mV s⁻¹), as pointed out by the pink arrow. **g, h** Full transmittance profiles (350–2250 nm) at 2.0–4.0 V (20 mV s⁻¹) and 1.5–4.0 V (10 mV s⁻¹), respectively. **i** Potential profiles during galvanostatic detrappingfor the two trapping states. **j** Current profiles during potentiostatic detrapping for the two trapping states.

during galvanostatic detrapping, coincident with the occurrence of the sharp increase of optical transmittance[33]. It should be pointed out that, for a-WO₃ samples, the deposition parameters were chosen in order to study the degradation and rejuvenation on a reasonable time scale. It shows that a-WO₃ thin films with low porosity possessed much slower degradation rate. As revealed in Supplementary Fig. 6, colored states of a denser WO₃ film degraded by only ~14% after 3000 cycles, where the a-WO₃ films used in this paper degrade by ~30% after 1000 cycles. The characteristics of ion trapping/detrapping processes for WO₃ thin films with different porosities are fully consistent. Therefore, it is conclusive that, ion trapping takes place in both porous and dense WO₃ films. In fact, the ion trapping and detrapping are found to prevail in all other cathodic electrochromic oxides as well, i.e., MoO₃[47], TiO₂[46], Nb₂O₅ and Ta₂O₅ as revealed in Supplementary Fig. 7.

**Origins of *shallow* traps**

We combine X-ray photoelectron spectroscopy (XPS), Raman spectroscopy and transmission electron microscopy (TEM) to depict the entire picture of ion trapping and detrapping processes. Evaluation of valence states by XPS is shown in Fig. 2a. a-WO₃ at the pristine state is first studied as a reference. Four characteristic peaks are noted from the W-4f spectra at this state (Fig. 2a, pristine state). Considering the spin-orbit splitting, the two paired peaks are assigned to W⁶⁺ and W⁵⁺

states with an absolute W⁶⁺ dominance[37,40,48] (Fig. 2b and Supplementary Table 1 present the proportions of each W state), which is in good agreements with the observations from others[41]. The O-1s spectra show two contributions with peaks at 530.5 and 532.1 eV, respectively. In addition to the main peak at 530.5 eV originating from matrix oxygen, the peak at 532.1 eV is assigned to oxygen atoms bonding to adsorbed impurities[49,50]. Naturally, no characteristic peak of Li-1s was detected in the pristine film.

We consider films working in the range of 2.0–4.0 V first, where *shallow* trap is the main cause for performance degradation. As Li ions are inserted (Fig. 2a, 5th colored state) to achieve dark blue, an obvious Li-1s signal can be seen at ~55 eV[47,49–52]. Simultaneously, W⁶⁺ are reduced to W⁵⁺ as a result of electron insertion from the external circuit, and optical absorption is achieved due to small *polaron* hopping between W⁵⁺ ↔ W⁶⁺ sites[15,21], as illustrated in Eq. 1. Because of the low photo-ionization cross-section of Li in XPS, the *x* values in Li$_x$WO₃ were calculated here by different proportions of W valences, instead of deriving it directly from Li-1s signals. *x* is found to be 0.43 at this stage, this is slightly larger than 0.35 which is believed to serve as a 'rule of thumb' for safe long-term device operation[6]. Once Li ions are released through a reverse scan (i.e., from 2.0 V to 4.0 V), optical transmittance returns to its initial state and all XPS spectra are identical to the reference (Fig. 2a, 5th bleached state). At this stage, the ion trapping

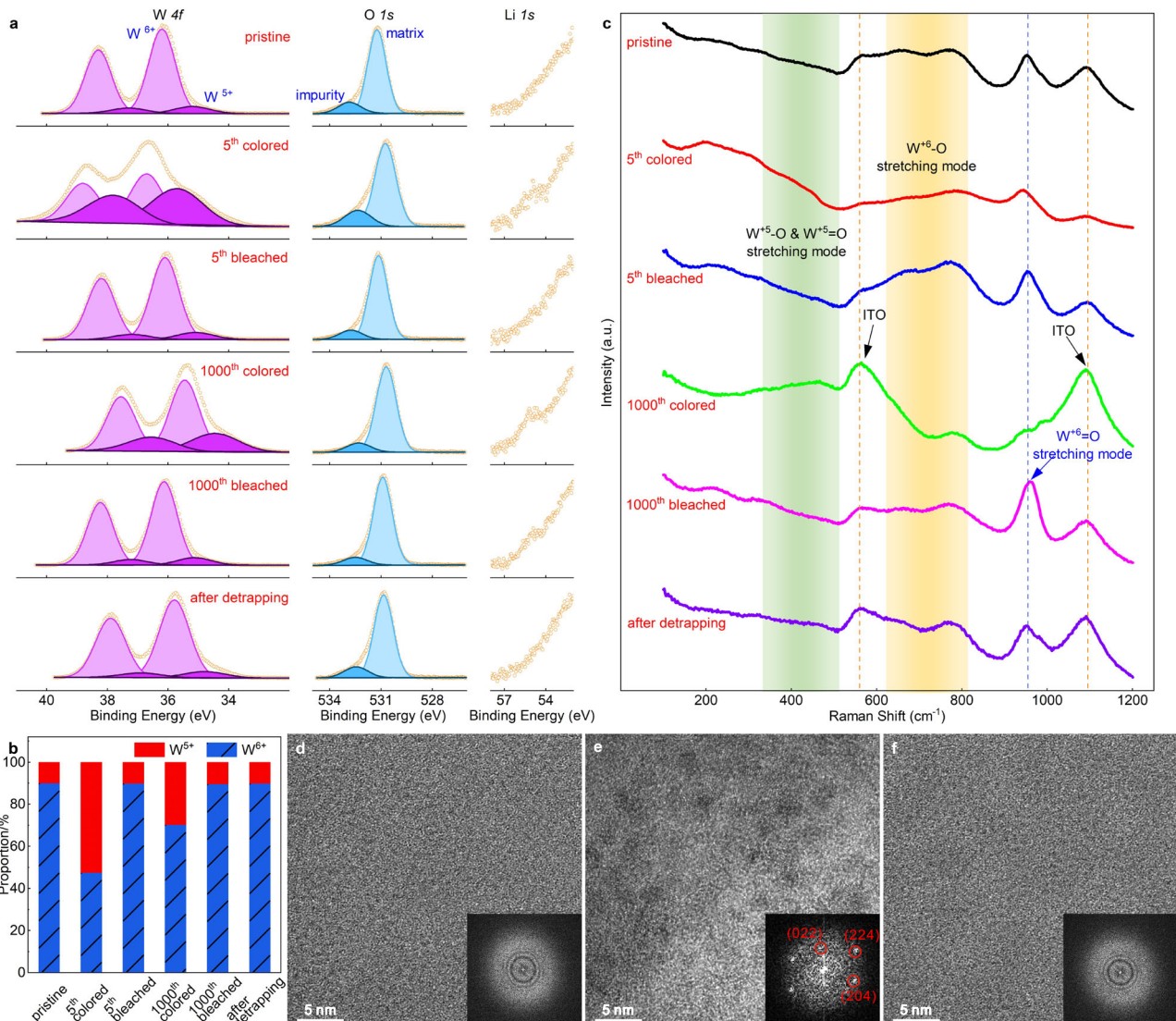

**Fig. 2 | Composition and structural characterization of *shallow* traps induced degradation (cycling in the range of 2.0–4.0 V). a** XPS characterization for $WO_3$ at various states and Pristine state is shown as a reference. Yellow circles represent the raw data, purple and blue lines stand for the signals from $W^{6+}$ and $W^{5+}$. The shaded regions by the lines indicate the portion of marked valency. **b** the proportions of each W valency. **c** Raman characterization for $WO_3$ at various states. Indium tin oxide, denotes as ITO, served as the substrate for film deposition. The two yellow dash lines represent the Raman signals from ITO. The blue dash line indicates the peak position for $W^{+6} = O$ stretching. Shaded green and yellow regions illustrate the $W^{+5}$-O and $W^{+5} = O$, as well as $W^{+6}$-O stretching. **d–f** HR-TEM images of $WO_3$ film at pristine state, 1000th bleached state and after detrapping, showing the formation of orthorhombic $Li_2WO_4$ upon long-term cycling and its release after detrapping. Insets: associated FFT patterns.

effect is not noticeable because of the limited number of cycles (only 5 CVs).

At the colored state of the 1000th cycle, the Li-*1s* signal increases (Fig. 2a, 1000th colored state). Note that the contribution to the Li-*1s* signal here is both from inserted Li ions in the 1000th cycle and accumulated Li ions in *shallow* traps over the past 1000 cycles. This statement is reasonable because the intensity of the $W^{5+}$ signal becomes weaker compared to the one at the 5th cycle, which is due to the lower amount of inserted Li ions at the 1000th cycle, as confirmed by the decreased reversible charge capacity in Fig. 1d. Additional evidence is that the intensity of Li-*1s* also largely decreases at the bleached state of the 1000th cycle while W-*4f* returns to its initial state (Fig. 2a, 1000th bleached state), indicating that the weak Li-*1s* intensity only arises from Li ions in *shallow* traps. Since the amount of $W^{5+}$ decreases, the efficiency of small *polaron* hopping between $W^{5+} \leftrightarrow W^{6+}$ sites drops (Supplementary Fig. 8), and thus optical modulation, i.e., $\Delta T$, is significantly reduced. The results shown here emphasize that

accumulated ions in *shallow* traps associated with pairing electrons do not reduce $W^{6+}$ to $W^{5+}$ or even $W^{4+}$ in the host matrix, suggesting the non-coloration nature of the *shallow* traps, therefore optical transmittance at the bleached state can always reach its initial value (Fig. 1c, g). In addition, no signal from Cl is detected (Supplementary Fig. 9) which means it is not involved into trapping processes. One may note that it shows a nonnegligible binding energy position shift of both $W^{6+}$ and $W^{5+}$ (as well as $W^{4+}$ in the next part in Fig. 3a) which was also observed from others[37,53]. We temporarily ascribe this slight shift to coordination environment variation upon ion insertion and extraction.

Raman spectra are shown in Fig. 2c, for a pristine film on ITO/glass (Fig. 2c, pristine state). Five broad peaks are noted whereas the peaks centered at ~560 $cm^{-1}$ and ~1100 $cm^{-1}$ are assigned to ITO/glass (Supplementary Fig. 10). Two peaks centered at ~665 $cm^{-1}$ and ~775 $cm^{-1}$ are assigned to $W^{+6}$-O stretching modes of $WO_3$[52,54] (also see Supplementary Fig. 11 for the Raman spectrum of commercial $WO_3$ powders with a grain size of ~20 nm and in hexagonal phase). The peak centered at

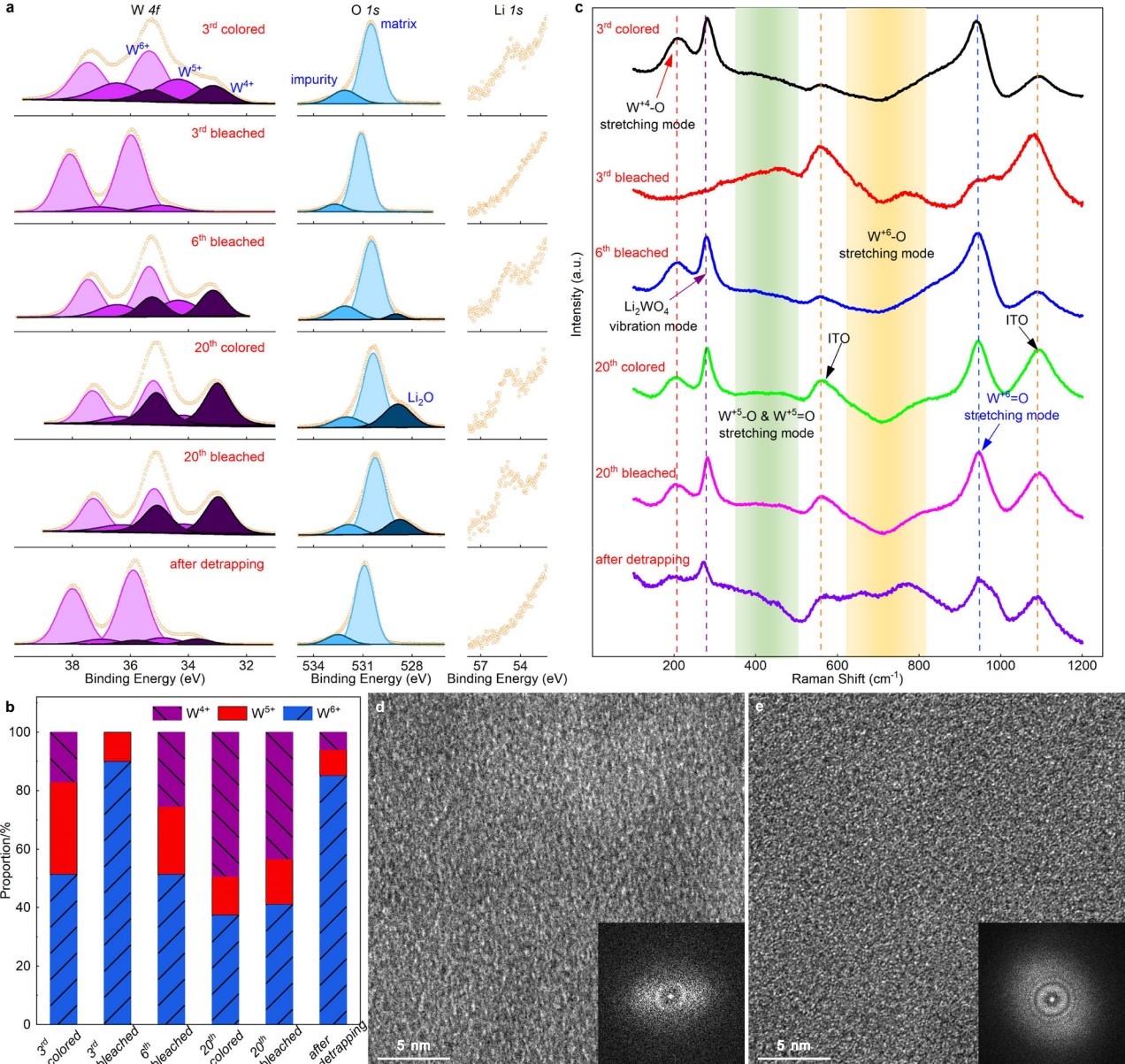

**Fig. 3 | Trapping and detrapping dynamics for WO₃ at 1.5–4.0 V. a** XPS data for $WO_3$ at various trapping/detrapping states. Yellow circles represent the raw data. Purple, blue and black lines, with associated shaded regions, stand for the signals from $W^{6+}$, $W^{5+}$ and $W^{4+}$, respectively. **b** the proportions of various W valences. **c** Raman data for $WO_3$ at various trapping/detrapping states. Indium tin oxide, denotes as ITO, served as the substrate for film deposition. The two yellow dash lines represent the Raman signals from ITO. The red, purple and blue dash lines indicate the peak position for $W^{+4}$-O stretching, $Li_2WO_4$ vibrating and $W^{+6}$ = O stretching, respectively. Shaded green and yellow regions illustrate the $W^{+5}$-O and $W^{+5}$ = O, as well as $W^{+6}$-O stretching. **d, e** HRTEM images of $WO_3$ at 3rd colored state and 20th bleached state, indicating the amorphous characteristics of $Li_2WO_4$ in these conditions. Insets: associated FFT patterns.

~950 cm⁻¹ is assigned to the stretching mode of $W^{+6}$ = O, which is characteristic for $a$-$WO_3$, arising from terminal oxygen atoms on the surfaces of clusters and/or microvoid structures[55,56]. When ions are inserted, the intensity of stretching modes due to $W^{+6}$-O and $W^{+6}$ = O clearly decreases (Fig. 2c, 5th colored state) while an enhanced intensity from ~330 cm⁻¹ to ~450 cm⁻¹ is noted because of the emerging stretching modes of $W^{+5}$-O and $W^{+5}$ = O[57]. This demonstrates that some of the $W^{+6}$-O and $W^{+6}$ = O bonds are broken and new $W^{+5}$-O and $W^{+5}$ = O bonds are formed due to the inserted Li ions. As expected, this is consistent with the XPS results that insertion of an ion-electron pair leads to a reduction of $W^{6+}$ to $W^{5+}$ in tungsten bronze ($Li_xWO_3$). One may note that the signal between ~330 cm⁻¹ and ~450 cm⁻¹ here is not sharp, this is because the increased fluorescence due to coloration[37]. The amorphous nature of the $WO_3$ electrode may also decrease peak intensity (Supplementary Fig. 12 for Raman measurements of $WO_3$ on

W/Si substrates to distinguish this disturbance). As ion/electron pairs are extracted, the features of the Raman spectra return to the ones analogous to the pristine state (Fig. 2c, 5th bleached state). After 1000 cycles, the intensity of $W^{+6}$-O and $W^{+6}$ = O stretching modes are enormously reduced at the colored state (Fig. 2c, 1000th colored state). Even after bleaching (Fig. 2c, 1000th bleached state), peak intensities are only partly recovered, suggesting that ions residing in *shallow* traps suppress the stretching of $W^{+6}$-O and $W^{+6}$ = O modes rather than reduce $W^{6+}$ to $W^{5+}$.

High-resolution TEM (HRTEM) and fast Fourier transform (FFT) patterns of a pristine sample indicate an amorphous nature of the as-deposited samples (Fig. 2d). At the bleached state of the 1000th cycle (Fig. 2e), nano crystalline grains embedded in the amorphous host are found and FFT patterns of these nanograins are assigned to orthorhombic $Li_2WO_4$ (JCPDS No: 28-0596). We investigated more than 10

samples after long-term cycling and the results are consistent and showing indisputable evidence that orthorhombic $Li_2WO_4$ were formed. Some representatives are provided in Supplementary Fig. 13. Grazing incidence XRD has also been performed on a severely aged $WO_3$ film (cycled in the range of 2.0–4.0 V) and confirmed the formation of orthorhombic $Li_2WO_4$ (Supplementary Fig. 14). After detrapping, these nanograins vanished as validated by both HR-TEM and the FFT pattern, leaving the amorphous phase of the electrode (Fig. 2f). Thicker $WO_3$ films (i.e., 800 nm) were also deposited and subjected to long term cycling (2000 CV cycles). TEM results show that the concentration of orthorhombic $Li_2WO_4$ nanograins has an obvious incremental gradient distribution from ITO substrate to film surface where the formed orthorhombic $Li_2WO_4$ nanograins are very rare near ITO substrate (Supplementary Fig. 15), suggesting orthorhombic $Li_2WO_4$ is more easily formed near the film surface.

As shown by spectroelectrochemistry in Fig. 1c, XPS in Fig. 2a and Raman spectroscopy in Fig. 2c, the initial performance is completely regained after detrapping. This suggests that orthorhombic $Li_2WO_4$ was effectively decomposed and Li ions were fully extracted from the host matrix. Bearing in mind that the $W^{6+}$ valency is maintained in orthorhombic $Li_2WO_4$, therefore the bleached state can always reach its initial value in every cycle upon long-term cycling (Fig. 1c, g). It should be pointed out that no signal was detected for orthorhombic $Li_2WO_4$ from Raman measurements (Fig. 2c), which is consistent with the observation reported in ref. 37. From the TEM analysis that the formed orthorhombic $Li_2WO_4$ nanograins were found small in quantity and rather dispersed in the amorphous host, as well as the amorphous host gives a strong background in Raman measurements, leading to the un-observed signal. However, the formed orthorhombic $Li_2WO_4$ accounts for the gradually degraded colored state due to: (i) formation of a $Li_2WO_4$ molecule is equivalent to consume a $WO_6$ octahedron which was used to achieve reversible coloration; (ii) dispersed $Li_2WO_4$ nanograins in an amorphous host, especially relatively higher concentration near the surface, also block the migration of Li-ions. A combination of the two contributions leads to a degradation of electrochromic performance.

So far, it can be concluded that formation of orthorhombic $Li_2WO_4$ during long-term cycling is the origin of *shallow* traps in *a*-$WO_3$ films. It should be noted that, in addition to the inserted Li ions from electrolyte, there is an extra oxygen atom in an orthorhombic $Li_2WO_4$ molecular, as compared to the molecular of a $WO_3$. Due to the incremental gradient distribution of the formed orthorhombic $Li_2WO_4$ nanograins, it suggests the extra oxygen is from the electrolyte (i.e., $LiClO_4$-PC). To validate this hypothesis, $LiClO_4$-PC was replaced by $LiPF_6$-PC and an accelerated degradation was observed (Supplementary Fig. 16a). XPS measurements showed gradient F signal was found for samples at the bleached state after long term cycling (Supplementary Fig. 16b, c). It is known that $F^−$ has a larger electronegativity than $O^{2−}$, refs. 58,59, which indicates $F^−$ can be easily incorporated into $WO_3$ than $O^{2−}$ and results in the accelerated degradation. Analogous to the films cycling in $LiClO_4$-PC that no-detected Cl signal, P signal was either found for films cycled in $LiPF_6$-PC (Supplementary Fig. 16d). The solvent of PC has also been replaced to EC/DEC (i.e., the new electrolyte is $LiClO_4$-EC/DEC) and yielded a consistent result as compared to $LiClO_4$-PC (Supplementary Fig. 17), suggesting the external oxygen is not from solvent. The above results reveal that, to build $WO_3$ based electrochromic devices, the selection of salt for the electrolyte should be more seriously considered than the solvent. The extra oxygen from ITO substrate was also excluded, as shown in Supplementary Fig. 18. Moreover, oxygen deficient tungsten oxide ($WO_{3-z}$) thin films, which appeared to be bluish in the pristine state due to the large number of oxygen vacancies induced $W^{5+}$ (Supplementary Fig. 19), were subjected to the same cycling as done for *a*-$WO_3$. The gradually increased transmittance at bleached states upon cycling was due to the oxidization from $W^{5+}$ to $W^{6+}$, which has to be achieved through oxygen

incorporation from electrolyte. Finally, the depth profile of oxygen (Supplementary Fig. 20) showed obvious decrease from film surface to ITO, indicating that oxygen is also from the electrolyte, and is in good accordance with, and also well explains, the TEM results which shows orthorhombic $Li_2WO_4$ nanograins concentrate in near the surface.

## Origin of *deep* and *irreversible* traps

Next, we consider the *deep* and *irreversible* traps, i.e., films cycled in the range 1.5–4.0 V (Fig. 3). At colored state of the 3rd cycle (Fig. 3a, 3rd colored state), a large amount of Li ions is inserted as indicated by the strong Li-*1s* signal. From W-*4f* spectra, three paired peaks are noted and this is different from the case of cycling in the range 2.0–4.0 V. The newly emerged pair of peaks are assigned to the spin-orbit splitting of $W^{4+}$ [40,48]. The formed $W^{4+}$ largely decreases the optical transmittance at the short-wavelength region as presented in Fig. 1h (see also Supplementary Fig. 4). This is in contrast to small *polaron* hopping of $W^{5+} \leftrightarrow W^{6+}$ transitions (i.e., Eq. 1), from which a broad transmittance peak in the short-wavelength region is shown at the colored state. Based on the W proportions, $x$ in $Li_xWO_3$ at this stage is calculated to be 0.56, larger than 0.43 at the 5th colored state in 2.0–4.0 V; this is in accordance with the larger charge capacity and stronger Li-*1s* signal at this state.

When a positive scan is applied for bleaching (Fig. 3a, 3rd bleached state), the valence states in the W-*4f* spectra return to their initial state and the Li-*1s* signal also disappears, suggesting a reversible intercalation at the current stage (i.e., ion trapping, if it occurs, is insignificant). At the 6th cycle bleached state (Fig. 3a, 6th bleached state), substantial ion trapping can be noted from the Li-*1s* spectra. The inserted electrons also reduce $W^{6+}$ to $W^{5+}$ and $W^{4+}$, as seen from the W-*4f* spectra. Meanwhile, a strong absorption can be observed in the short-wavelength region due to the strong increase of $W^{4+}$ sites (Supplementary Fig. 4, 1.5–4.0 V, 6th bleached). This is in striking contrast to the *shallow* traps (Fig. 2a, 1000th bleached state), where resided ions/electrons do not change the valence state of tungsten and thus the optical transmittance can always be identical to the initial value.

As shown in Fig. 1h, optical modulation at the 20th cycle is extremely faint and the intensity of the $W^{4+}$ and Li signals are further enhanced at both colored and bleached states (Fig. 3a, 20th colored state and 20th bleached state). In addition, a peak at 528.3 eV appears in the O-*1s* spectra, suggesting that formation of $Li_2O$ [50,60] takes place as cycling proceeds. In fact, a trace of $Li_2O$ is already noticed at the bleached state of the 6th cycle (Fig. 3a, 6th bleached state).

The increased proportion of $W^{4+}$ at these states (Fig. 3b and Supplementary Table 2) yields an enhanced optical absorption in the short-wavelength region, thus confirming that optical absorption at short wavelengths originates from bipolaron hopping between $W^{4+} \leftrightarrow W^{6+}$ sites. Here, small polaron hopping between $W^{4+} \leftrightarrow W^{5+}$ sites is unlikely, because $W^{5+}$ gradually vanishes during continuous ion insertion (Fig. 3b and Supplementary Table 2), while an increased/decreased optical absorption at short-/long-wavelength regions in the full spectrum is simultaneously observed (Supplementary Fig. 4). This is typical for a combination of decreased/increased hopping by $W^{5+} \leftrightarrow W^{6+}$ and $W^{4+} \leftrightarrow W^{6+}$ transitions (switching of the two modes can be seen from Supplementary Movie 1). The optical absorption in the short-wavelength region also suggests that a higher energy is needed to activate $W^{4+} \leftrightarrow W^{6+}$ hopping. Therefore, in addition to the polaron hopping mode as described in Eq. 1, a new hopping mode between $W^{4+} \leftrightarrow W^{6+}$ also exist and can be written as:

$$W_{(a)}^{4+} + W_{(b)}^{6+} \overset{h\nu_2}{\leftrightarrow} W_{(b)}^{6+} + W_{(b)}^{4+} \qquad (2)$$

When the severely aged film has gone through a galvanostatic detrapping process, XPS signals for both Li and $Li_2O$ disappear (Fig. 3a, after detrapping). However, a weak signal for $W^{4+}$ remains which is consistent with the observation of *irreversible* traps that are not

recoverable and result in optical absorption at short wavelengths as presented in Fig. 1h. It should be noted that $W^{4+}$ sites cannot be formed without inserted Li ions, thus the un-observed Li signal here is only due to the low concentration of Li ions as well as the detection limit for light elements in XPS.

As illustrated in Fig. 3c of the Raman spectra, the stretching modes of $W^{+6}$-O and $W^{+6}$ = O in $a$-$WO_3$ are already severely inhibited at the colored state in the 3rd cycle (Fig. 3c, 3rd colored state). This is due to the larger number of inserted ions/electrons, as compared to the case of 2.0–4.0 V (see Fig. 1d, e). More importantly, two more peaks emerge: one centered at ~220 $cm^{-1}$ due to stretching modes of $W^{+4}$-O[56] which is well consistent with the observation of $W^{4+}$ signals from the XPS measurements (Fig. 3a, 3rd colored state); the other centered at ~282 $cm^{-1}$ and due to $Li_2WO_4$[37,61]. The HR-TEM images and the diffuse rings in the FFT pattern (Fig. 3d) demonstrate that the structure is still discorded at the colored state of the 3rd cycle, indicating that the formed $Li_2WO_4$ is amorphous and mixed with the $a$-$WO_3$ host matrix. As the film is bleached (Fig. 3c, 3rd bleached state), peaks for stretching modes of $W^{+6}$-O and $W^{+6}$ = O are recovered, while the peaks assigned to $W^{+4}$-O and amorphous $Li_2WO_4$ vanished, suggesting that both $W^{4+}$ and $Li_2WO_4$ are reversible at this stage because of the limited number of cycles. Here the appearance and disappearance of $W^{+4}$-O and $Li_2WO_4$ are concomitant, suggesting a coupling between $W^{4+}$ and $Li_2WO_4$. This is rational, because there is an extra oxygen atom in $Li_2WO_4$ as compared to the initial $WO_3$, and one of the three oxygen atoms which originally binds to $W_{(a)}^{6+}$ migrates to a nearby $W_{(b)}O_3$ unit and forms $Li_2W_{(b)}O_4$, leaving the $W_{(a)}$ atom to be reduced to $W_{(a)}^{4+}$ (Fig. 4a, blue dash-line square). This configuration is in agreement with our experimental observations of the $W^{4+}$-$Li_2WO_4$ coupling and explains well the origin of the extra oxygen atom in coupled $W^{4+}$-$Li_2WO_4$. As noted, this is different from the case for long-term cycling in the range of 2.0–4.0 V, where the external oxygen is gradually incorporated from electrolyte.

The bleached state at the 6th cycle, as well as colored and bleached states at the 20th cycle are discussed together, since little difference is noted among these three states (Fig. 3c, 6th bleached, 20th colored and 20th bleached). In these states, peaks for $W^{+6}$-O and $W^{+6}$ = O stretching modes are enormously inhibited while peaks for $W^{4+}$-O and amorphous $Li_2WO_4$ are rather intense. This indicates that films at these states are severely degraded and a limited number of active sites are available for coloration, thus the films almost lose their optical modulation, as confirmed by the in-situ spectro-electrochemistry results in Fig. 1e, h. On the other hand, no peak of $Li_2O$ in the Raman spectra is detected, suggesting the formation of $Li_2O$ is limited to the film surface and is small in quantity, which is also confirmed by XPS depth profile (Supplementary Fig. 21). The $Li_2O$ on the surface is also in accordance with its sequence of formation and decomposition, as will be shown later.

Once the detrapping is conducted, peaks for the stretching modes of $W^{+6}$-O and $W^{+6}$ = O are approximately recovered, suggesting a quasi-complete rejuvenation of the film. Besides, signals at ~220 $cm^{-1}$ and ~282 $cm^{-1}$, although weak, are still present, indicating that part of the coupled $W^{4+}$-$Li_2WO_4$ can't be decomposed any longer. The associated optical transmittance in the short-wavelength region is not completely recovered (Fig. 1h). We therefore conclude that the origin of *irreversible* traps is non-decomposed $W^{4+}$-$Li_2WO_4$. In the XPS spectra, a trace of $Li_2O$ is noticed at the 6th cycle and this intensity strongly increases at the 20th cycle. After detrapping, the signal assigned to $Li_2O$ vanished first. In fact, formation and decomposition of $Li_2O$ is coupled to $W^{4+}$, which will be justified in the next section in order to clarify the origin of *deep* traps.

## Capturing the detrapping dynamics of *deep* traps

So far, the detrapping dynamics for the *deep* traps is still unknown, and this will be revealed herein. For severely aged films in the range of 1.5–4.0 V, the optical transmittance shows a three-step characteristic upon galvanostatic detrapping. Five points are selected, as marked in Fig. 4b, to elaborately uncover the associated dynamics. Point *I* was selected at the middle of the first milder transmittance increase. A slight drop of $W^{4+}$ and increase of $W^{5+}$ intensity is noted (Fig. 4c, Point *I* state) when comparing to the *deep* trapped state (proportions of each W state can be seen in Fig. 4d and Supplementary Table 3). This indicates that the slightly increased optical transmittance is due to partial oxidation of $W^{4+}$ upon detrapping. Thus, the small *polaron* hopping efficiency involving $W^{4+}$ is slightly declined, yielding an increased optical transmittance. On the other hand, the peak intensity for amorphous $Li_2WO_4$ in the Raman spectra is hardly varied (Fig. 4e, Point *I* state), suggesting that the present release of $W^{4+}$ is independent of $Li_2WO_4$. From the XPS spectra, it can be found that the signal assigned to $Li_2O$ vanished at stage *II* (Fig. 4c, Point *II* state), and this indicates that the $W^{4+}$ released here is coupled with $Li_2O$. In fact, the formation $Li_2O$ consumes an oxygen atom from a $WO_6$ octahedron and induces a $W^{4+}$ formation simultaneously as shown in Fig. 4a (purple dash-line square).

Points *II*, *III* and *IV* are selected to be just after the first milder transmittance increase, at the middle of the plateau, and at the turning point prior to the second transmittance increase, respectively. From point *II* to *IV*, the intensity of W-$4f$ peaks show almost no variation whereas Li-$1s$ shows a consecutive reduction until it vanished. Note that the vanished Li-$1s$ signal here is due to the low concentration of Li ions at this state, considering the detection limit for light elements of XPS. Consistently, Raman spectra (Fig. 4e, point *II*, *III* and *IV* states) show that stretching modes of $W^{+6}$-O and $W^{+6}$ = O are steadily enhanced because of ion release. Therefore, the combination of un-varied W-$4f$ signal, the decreased intensity of Li-$1s$ in XPS and the recovered stretching modes of $W^{+6}$-O and $W^{+6}$ = O in Raman spectra suggest that the ion release is from decomposition of amorphous $Li_2WO_4$ without varying the valence state of W. Besides, when cycled at 2.0–4.0 V after point *IV*, both CV curves and optical modulations show notable rejuvenation (Supplementary Fig. 22), again providing a strong proof for this claim.

Point *V* was selected to be at the middle of second transmittance increase where the transmittance is largely recovered. It can be noted that the peak intensity for $W^{4+}$ is greatly reduced in both XPS and Raman spectra, indicating a decomposition of the coupled $W^{4+}$-$Li_2WO_4$. This process is very rapid and different from the situation for *shallow* traps. Because of the rapid decomposition of coupled $W^{4+}$-$Li_2WO_4$, the current density shows a peak simultaneously with the occurrence of the sharply increased transmittance (Fig. 4b). As observed in Fig. 4e (Point *V* state), when the detrapping process is finished, the Raman spectra return to the pristine state, expect for the coupled $W^{4+}$-$Li_2WO_4$ from *irreversible* traps. One may notice that the peak at ~282 $cm^{-1}$ seems to be increased, but this is because of vanished fluorescence vanishes due to the bleaching of the film after detrapping, thus reducing the background intensity around this range (Supplementary Fig. 12 for Raman measurements of $WO_3$ on W/Si substrates to distinguish this disturbance).

Now, we discuss the different forms of $W^{4+}$ sites. When conducting potentiostatic insertion of Li ions at 2.0 V for one minute, $W^{4+}$ is formed (Fig. 4c, e, 2.0 V, 1.0 min state). However, no signal from $Li_2WO_4$ or $Li_2O$ is found in XPS and Raman spectra, suggesting that $W^{4+}$ forms in the same fashion as in tungsten bronze ($Li_xWO_3$), as shown in Fig. 4a (pink dash-line square). We term the current $W^{4+}$ sites as type *I*. Type *I* forms first and, more importantly, they are reversible, as proved by a complete ion extraction at 4.0 V, as shown in Supplementary Fig. 23. As Li ions are continuously inserted at 2.0 V or by expanding the low potential limit to 1.5 V, the host matrix gets reconfigured to form coupled $W^{4+}$-$Li_2WO_4$ (type *II*), as indicated in Fig. 4c, e (2.0 V, 20 h) and Fig. 3c. Type *II* can be reversible, releasable or irreversible as we discussed in the context of Fig. 3, and this is likely dependent on the

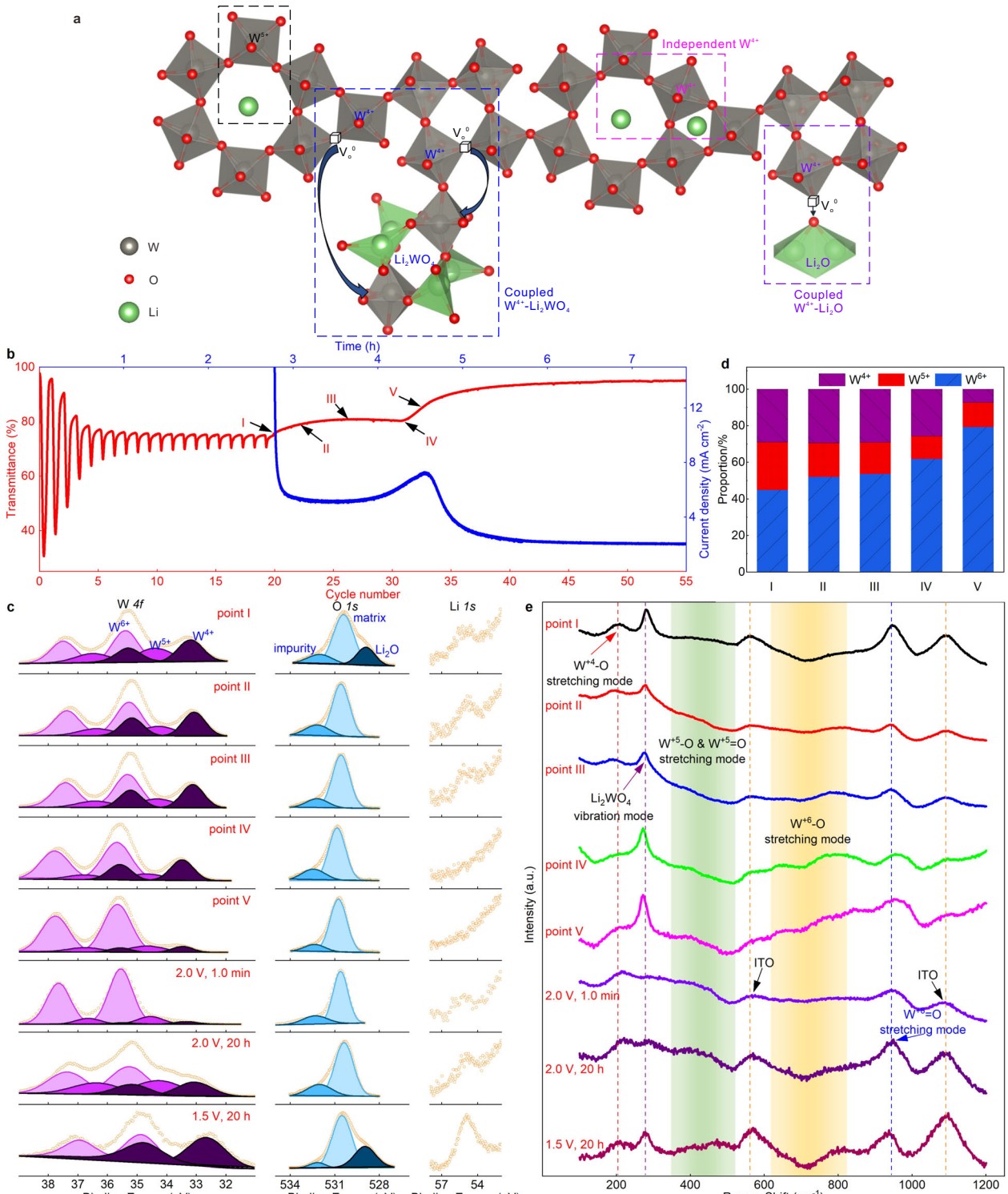

**Fig. 4 | Various states of WO₃ during Li ion intercalation, trapping and detrapping. a** Schematic illustration of the formation of independent $W^{5+}$ and $W^{4+}$, coupled $W^{4+}$-$Li_2WO_4$ and coupled $W^{4+}$-$Li_2O$ in amorphous $WO_3$ host matrix. The arrows illustrate the possible migrate paths for oxygen. The white cubes, denote as $V_o^0$, indicate the oxygen vacancies. **b** In-situ optical transmittance during the ion trapping and detrapping process. The current density profile upon potentiostatic detrapping shows a peak when the sharp increase of the optical transmittance occurs. Insets mark the 5 selected points to be tested by XPS and Raman spectroscopy. **c** XPS results for the selected points as well as during potentiostatic ion insertion. Yellow circles represent the raw data. Purple, blue and black lines, with associated shaded regions, stand for the signals from $W^{6+}$, $W^{5+}$ and $W^{4+}$, respectively. **d** Proportions of each W state of the selected points. **e** Raman results for the selected points as well as during potentiostatic ion insertion. Indium tin oxide, denotes as ITO, served as the substrate for film deposition. The two yellow dash lines represent the Raman signals from ITO; the red, purple and blue dash lines indicate the peak position for $W^{+4}$-O stretching, $Li_2WO_4$ vibrating and $W^{+6}$ = O stretching, respectively. Shaded green and yellow regions illustrate the $W^{+5}$-O and $W^{+5}$ = O, as well as $W^{+6}$-O stretching.

associated configurations. Finally, coupled $W^{4+}$-$Li_2O$ (type *III*) is formed after the generation of $W^{4+}$-$Li_2WO_4$, and decomposed at the very beginning of the detrapping process. In fact, $Li_2O$ can even be decomposed at 4.0 V, as confirmed by the release of coupled $W^{4+}$-$Li_2O$ after potentiostatic detrapping at 4.0 V (Supplementary Fig. 24). Overall, $W^{4+}$, in any form, leads to optical absorption in the short-wavelength region due to the emergence of $W^{4+} \leftrightarrow W^{6+}$ hopping. The spectroelectrochemistry, XPS, and Raman results support the fact that decomposition of these three forms of $W^{4+}$ is just the opposite process to their formation. However, a small quantity of coupled $W^{4+}$-$Li_2WO_4$ units are not decomposable are described as *irreversible* traps.

Based on the above discussion and experimental evidence, Fig. 5 provides a general picture of the ion intercalation, trapping and detrapping in *a*-$WO_3$. Optical absorption in tungsten bronze originates from small polaron hopping $W^{5+} \leftrightarrow W^{6+}$ transitions, which results in optical absorption in the whole range, especially in the long-wavelength region (i.e., 550 nm–2250 nm). $W^{4+}$ can also form in tungsten bronze depending on the degree of ion insertion, and associated $W^{4+} \leftrightarrow W^{6+}$ transitions lead to an optical absorption in the short-wavelength region. After long-term cycling, orthorhombic $Li_2WO_4$ is formed in which $W^{6+}$ sites are maintained, but the reversible sites for intercalation are largely reduced. Therefore, *polaron* hopping efficiency between $W^{5+} \leftrightarrow W^{6+}$ drops, yielding a diminished optical modulation. Coupled $W^{4+}$-$Li_2WO_4$, amorphous $Li_2WO_4$ and $W^{4+}$-$Li_2O$ can also form and their reversibility depends on the potential window as well as the extent of ion trapping. Some coupled $W^{4+}$-$Li_2WO_4$ could even not be decomposed and are denoted as *irreversible* traps. The origin of *deep* traps is complex, where ion trapping is a multiple-step-determined process, and is composed of the formation of $W^{4+}$-$Li_2WO_4$, amorphous $Li_2WO_4$ and $W^{4+}$-$Li_2O$. The ions residing in *shallow* and *deep* traps can be fully released through either a potentiostatic or galvanostatic detrapping process, therefore initial electrochromic performance can be completely regained. Another difference between shallow and deep traps is that, disproportionation reaction takes places to form coupled $W^{4+}$-$Li_2WO_4$ in deep traps and extra oxygen is from the adjacent oxygen in the host matrix rather than electrolyte.

It should be pointed out that the origin of *shallow* traps was never touched upon, due to the unvarying bleached state. Some hints of *deep* traps were indicated in previous research work. Hashimoto et al.[32] subjected *a*-$WO_3$ to heavy lithiation in an electrolyte containing 3000 ppm water, and observed that $7Li_2WO_4 \cdot 4H_2O$ was formed in the colored state where the film showed a black-brown color. It was proposed that $W^{4+}$ was responsible for the black-brown color. Indeed, $W^{4+}$ is the main cause of this color, i.e., by the strong absorption at the short-wavelength region, but it bonds to a more complex structure as we revealed above. Takayanagi et al.[37] conducted in-situ hard XPS also on heavily lithiated *a*-$WO_3$ and proposed that formation of $Li_2WO_4$ eroded the electrochromic performance. However, direct identification of $Li_2WO_4$ was missing. Cao et al.[39] directly observed formation of $Li_2WO_4$ through in-situ TEM in heavily lithiated hexagonal $WO_3$ nanowires. However, no long-term cycling was conducted in both Takayanagi's and Cao's work, which makes it hard to connect the formed $Li_2WO_4$ to various traps and the associated electrochromic performance.

With a combination of in-situ spectroelectrochemistry and other ex-situ spectroscopic techniques, the origin of the three different types of traps are demonstrated, and the ion trapping and detrapping dynamics as well as coloration mechanisms in *a*-$WO_3$ thin films is provided. Ion trapping is also found to take place in films with low porosity, although the degradation rate is relatively slower. The origin of *shallow* traps is orthorhombic $Li_2WO_4$ which formed during long-term cycling in the range 2.0–4.0 V. *Shallow* traps degrade the colored state by immobilizing reversible intercalation sites and block migration channels for Li-ions. As the potential window is expanded to 1.5–4.0 V, *deep* traps formed which degraded both the bleached and colored states. Importantly, *deep* traps are found to be more complex, composed of coupled $W^{4+}$-$Li_2WO_4$, amorphous $Li_2WO_4$ and coupled $W^{4+}$-$Li_2O$. The detrapping process shows that ion release from *deep* traps is sequential, where decomposition of $W^{4+}$-$Li_2O$ occurs first, followed by decomposition of amorphous $Li_2WO_4$. Decomposition of coupled $W^{4+}$-$Li_2WO_4$ takes place last, and part of it is unable to be decomposed which accounts for the *irreversible* traps. In addition, three forms of $W^{4+}$ are found in *a*-$WO_3$ whose reversibility is associated with the coupled species. We also confirm that polaron hopping due to $W^{6+} \leftrightarrow W^{4+}$ transitions results in optical absorption at the short-wavelength region. Our findings demonstrated in this paper provide a general picture of ion reversible intercalation, trapping and detrapping in a model electrochromic material, $WO_3$, which is consultative to other materials and devices of ion intercalation based. For example, ion trapping induced degradation was also found in other cathodic oxides (i.e., $Nb_2O_5$, $Ta_2O_5$, $MoO_3$ and $TiO_2$) whereas ion detrapping was proved to be effective. With the understanding of trapping/detrapping dynamics, this work may pave a way for future attempts to suppress or even eliminate traps with the aim to develop superior electrochromic devices. Moreover, the methodologies employed here can be well expanded to other electrochromic materials.

## Methods

### Materials

All the materials are available commercially and used as received. The target was a 3-inch-diameter plate of metallic tungsten/niobium/tantalum (99.999%). Argon and oxygen gas are both with 99.998% purity. Clean $In_2O_3$: Sn (ITO) coating glasses with a resistance of 60–80 Ω per square are used as the substrate. Lithium foil (Alfa Aesar, 99.9%), propylene carbonate (PC, Sigma-Aldrich, 99.7%), Diethyl carbonate (DEC, Aladdin, 99%), Ethylene carbonate (EC, Aladdin, 99%), lithium perchlorate ($LiClO_4$, Aladdin, 99.99%), lithium hexafluorophosphate ($LiPF_6$, RHAWN, 99.5%) were utilized without initial treatments.

### Thin film deposition

Thin films of $WO_3$, $Nb_2O_5$ and $Ta_2O_5$ were deposited by reactive DC magnetron sputtering. The distance between target and substrate is ~10 cm. The substrates were not intentionally heated during the film deposition. The deposition chamber was first evacuated to ~$6 \times 10^{-5}$ Pa. Pre-sputtering took place in argon (99.998%) and oxygen (99.998%) for 5 min prior to sample deposition. The power to the target was set at 200 W. For $WO_3$, the total pressure during sputtering was maintained at 4.0 Pa, $O_2$/Ar gas-flow was kept at a constant value of 5/35 sccm; for $Nb_2O_5$, the total pressure was 1.0 Pa, $O_2$/Ar gas-flow was

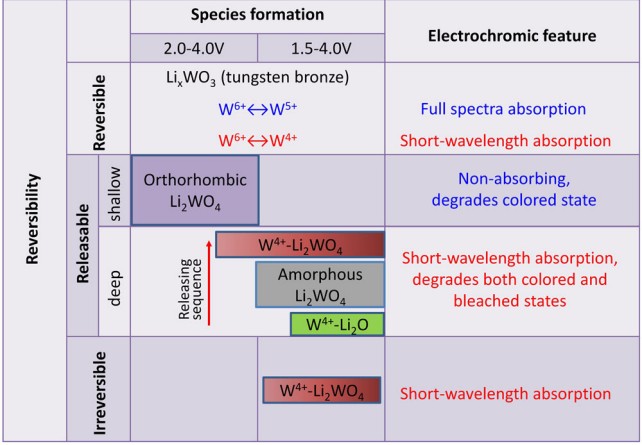

**Fig. 5 | A general picture of the ion intercalation, trapping and detrapping in *a*-$WO_3$.** It shows the formed species, reversibility and how it varies the optical modulation.

5/45 sccm; for $Ta_2O_5$, the total pressure was 4.0 Pa, $O_2$/Ar gas-flow was 5/30 sccm. The present conditions were chosen in order to produce porous films and to study degradation and rejuvenation on a reasonable time scale. As for the dense $WO_3$ films, the pressure was 1.0 Pa, with all the other parameters unchanged. The substrate is rotated during sputtering to ensure uniform coating. The thickness of the prepared films we used in this paper are about $300 \pm 30$ nm, which can be varied by changing the deposition time.

## Electrochromic and electrochemical measurements

The electrochemical and electrochromic measurements were carried out in an argon glove box with water/ oxygen content below $\sim 0.01$ ppm., using a custom-made spectroelectrochemical cell in the three-electrode configuration. The $WO_3$, $Nb_2O_5$ and $Ta_2O_5$ film served as the working electrode and was electrochemically cycled in 1 M $LiClO_4$ dissolved in propylene carbonate (or 1 M $LiPF_6$ in PC, or 1 M $LiClO_4$ in EC/DEC). Both the counter and reference electrodes were Li foils. Cycle voltammetry (CV), chronoamperometry (CA) and chronopotentiometry (CP) were conducted by an electrochemical workstation (Bio-logic, model VSP). For the CV measurement, the sweep rate is 20 mV s$^{-1}$ within 2.0–4.0 V to yield shallow trapped state, and 10 mV s$^{-1}$ within 1.5–4.0 V to yield deep trapped state, respectively. Charge capacity $Q$ (in units of mC cm$^{-2}$) was determined from cyclic voltammetry data by

$$Q = \int \frac{jdV}{s}$$

where $j$ is current density (in mA cm$^{-2}$), $s$ is the sweep rate (in V s$^{-1}$), and V is the voltage. Both CA and CP techniques were used to extract Li$^+$ ions from the films; specifically, a constant potential of 5.8 V, or a constant current of 3 µA cm$^{-2}$ was applied in the direction opposite to the one yielding Li$^+$-ion insertion in the host material. The corresponding potential level (5.5 V or more) will lead to electrolyte degradation, which accounts for a large amount of the charge passed. Depending on the purpose of the measurement, a sequence of CV, CA or CP with certain orders were programed to operate. In situ optical transmittance was recorded by using a fiber-optical instrument from Ocean Optics (QEpro, Ocean Optics, USA). The spectral range was 350–2250 nm, and optical data were recorded simultaneously with the electrochemical measurement. The sample is positioned in a quartz electrochemical cell, between a tungsten halogen lamp and the detector. The 100% level was taken as the transmittance recorded before immersion of the sample into the quartz cell with electrolyte. Both single wavelength data and full range (350–2250 nm) spectra were recorded.

## Materials characterization

Thickness was measured by using a AlphaStep D-300 from KLA Corporation, USA. X-ray diffraction (XRD) was performed on an X-ray diffractometer (Rigaku Smartlab) with the incident radiation at 40 kV and 200 mA using Cu K$\alpha$ ($\lambda = 1.5418$ Å) radiation. The Selected area electron diffraction (SAED) pattern of the films was examined using a transmission electron microscope (TEM; Tecnai F30, FEI, USA) operated at 300 kV. The TEM samples were prepared using focused ion beam (FIB; Dual Beam Heilos Nanolab 600i, FEI) milling to be 50–100 nm in thickness. Carbon is coated on the sample to protect sample before milling. XPS analysis was conducted using a Escalab Xi+ from Thermo Fisher Scientific with Al K$\alpha$ radiation, h$\upsilon = 1486.7$ eV. Raman spectroscopy was conducted using a LabRAM HR spectrometer. Spectra were acquired using a 532 nm laser at constant power (25%). All the tested samples were taken out of the electrolyte cell, and washed with dimethyl carbonate (DMC, Sigma-Aldrich, >99.5%) to rinse off the residual electrolyte on the surface. The rinsing and transfer process were conducted in the glove box with oxygen and water both less than 0.01 ppm. The sample was transferred to the XPS device through the vacuum transfer chamber, during this process, water and oxygen will not interfere with the sample.

## Reporting summary

Further information on research design is available in the Nature Portfolio Reporting Summary linked to this article.

## Data availability

The data used in this study are provided in the Source Data file. Source data are provided with this paper.

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

## Acknowledgements

This work is also financially supported by National Natural Science Foundation of China (No.52172294), the "Shenzhen Science and Technology Innovation Commission" (Grant No.JCYJ20210324105402007 and 20220815095607001), Guangdong Provincial Innovation and

Entrepreneurship Project (No. 2017ZT07C071), Department of Education of Guangdong Province (2023ZDZX3026), Guangdong-Hong Kong-Macau Joint Laboratory on Micro-Nano Manufacturing Technology (2021LSYS004), and National Natural Science Foundation of Guangdong (No. 4685326) and Guangdong Provincial Key Laboratory Program (2021B1212040001) from the Department of Science and Technology of Guangdong Province. The film deposition work used the resources from SUSTech Core Research Facilities that receives support from the Shenzhen Municipality.

## Author contributions

R.-T.W. initialized the project and supervised the work. R.Z. and R.-T.W. designed the experiments. R.Z. prepared the samples, conducted the characterizations and electrochemical measurements with assistance of S.H. and Q.Z. Y.Z. contributed to the TEM sample preparation and analysis. All authors co-wrote the manuscript. All authors discussed the results and commented on the manuscript.

## Competing interests

The authors declare no competing interests.
