## [Peer Review File · Nature Communications]

This manuscript has been previously reviewed at another journal that is not operating a transparent peer review scheme. This document only contains reviewer comments and rebuttal letters for versions considered at *Nature Communications*.

REVIEWER COMMENTS

Reviewer #1 (Remarks to the Author):

The authors report their analysis of lithium ion trapping and detrapping behavior in amorphous WO₃-based electrochromic devices. The ion trapping and detrapping has been a key issue which largely unknown to date and is closely related to degradation of the electrochromic device. Many researchers have thus been intensively investigating the mechanism. Although previous works, including ones by one of the present authors, indicated that some types of trapping sites causes degradation of the electrochromic function, the detail of the degradation, in particular the dynamics, has been unclear to date. In the present work, the present authors unveiled the lithium ion trapping and detrapping dynamics in the long-term cycling operation process of electrochromic window. Their results clearly demonstrate that there are different optical degradation modes, which are induced by shallow and deep traps. They clarified that the origin of the shallow traps and deep traps are orthorhombic Li₂WO₄ formation and W⁴⁺-Li₂WO₄ couples respectively. Furthermore, they evidenced that optical absorption of specific wavelength region is correlated with difference in the polaron hopping sites, between W⁵⁺-W⁶⁺ sites or W⁴⁺-W⁶⁺ sites, on the basis of various spectroscopic and microscopic studies. They succeeded to give significant advance in the understanding of electrochromic devices and the degradation/rejuvenation mechanism. Their findings are of great importance in development of next generation electrochromic windows with superior characteristics including long-term stability and cyclability. Their experimental and analysis methods are valid and their interpretation are reasonable. The conclusion of the work is supported by rigid experimental results and thus is conclusive. Furthermore, all of my technical comments in the previous review has been appropriately addressed in the present version. Therefore, the reviewer recommends publication of this work in the Nature Communications.

Reviewer: Dr Takashi Tsuchiya

Reviewer #2 (Remarks to the Author):

The authors have improved the manuscript and it may now be suitable for publication.

Reviewer #3 (Remarks to the Author):

1. The paper is lack of significant new results to support it to be published on nature communication.
2. WO₃ is a very typical electrochromic materials and most thin film fabrication methods can prepare a pretty high performance WO₃ thin film which can carry out electrochromic cycling for more than 10000 times, even 100000 times, without any degradation. The authors seem to deliberately use a higher voltage power 2.0-4.0 to degrade the WO₃. Only 5th or/and 20th cycles making the WO₃ degradation can not be equivalent to the real thousands cycles degradation. In addition, Li⁺ trapping problems is not very serious in WO₃ as like in NiOx.
3. One of the authors has published a series of papers on the trapping problems (Ref. 25, 30-32) around 2015. In this paper we do not see many new findings comparing with the old publications.
4. In figure 2 a, it shows a clear binding energy position shift of both W⁶⁺ and W⁵⁺, but there is no explanation of this shift found in the context.
5. The results from the measurement methods of XPS, Raman, TEM, spectral transmittance etc are not sufficient enough to support many of the statements in the paper such as the analysis and direct evidence of the three trapping modes.
6. In Figure 2a, the shown Li 1s dot signals can not see any signal peaks or any differences among all the lines except the 1000th colored state with a very small peak. Therefore the figure is non corresponding to the statements describing the changes of Li 1s signals.

RESPONSE TO REVIEWERS' COMMENTS

Reviewer #1:

General comments: The authors report their analysis of lithium ion trapping and detrapping behavior in amorphous WO₃-based electrochromic devices. The ion trapping and detrapping has been a key issue which largely unknown to date and is closely related to degradation of the electrochromic device. Many researchers have thus been intensively investigating the mechanism. Although previous works, including ones by one of the present authors, indicated that some types of trapping sites causes degradation of the electrochromic function, the detail of the degradation, in particular the dynamics, has been unclear to date. In the present work, the present authors unveiled the lithium ion trapping and detrapping dynamics in the long-term cycling operation process of electrochromic window. Their results clearly demonstrate that there are different optical degradation modes, which are induced by shallow and deep traps. They clarified that the origin of the shallow traps and deep traps are orthorhombic Li₂WO₄ formation and W⁴⁺-Li₂WO₄ couples respectively. Furthermore, they evidenced that optical absorption of specific wavelength region is correlated with difference in the polaron hopping sites, between W⁵⁺-W⁶⁺ sites or W⁴⁺-W⁶⁺ sites, on the basis of various spectroscopic and microscopic studies. They succeeded to give significant advance in the understanding of electrochromic devices and the degradation/rejuvenation mechanism. Their findings are of great importance in development of next generation electrochromic windows with superior characteristics including long-term stability and cyclability. Their experimental and analysis methods are valid and their interpretation are reasonable. The conclusion of the work is supported by rigid experimental results and thus is conclusive. Furthermore, all of my technical comments in the previous review has been appropriately addressed in the present version. Therefore, the reviewer recommends publication of this work in the Nature Communications.

Response: We appreciate for the reviewer's very positive comments that "*They succeeded to give significant advance in the understanding of electrochromic devices and the degradation/rejuvenation mechanism*", and "*Their findings are of great importance in development of next generation electrochromic windows with superior characteristics including long-term stability and cyclability*", as well as the reviewer's recommendation for publication of this work in *Nature Communications*.

Reviewer #2:

General comments: The authors have improved the manuscript and it may now be suitable for publication.

Response: We appreciate for the reviewer's approval for its publication in *Nature Communications*.

Reviewer #3:

Comment 1: The paper is lack of significant new results to support it to be published on nature communication.

Response: We appreciate the reviewer' comments. Nevertheless, we respectfully hold a different

perspective regarding the evaluation of the manuscript's novelty. Ion trapping and detrapping are crucial for the durability of electrochromic materials and devices, as also pointed out by reviewer #1 that “The ion trapping and detrapping has been a key issue which largely unknown to date and is closely related to degradation of the electrochromic device”. Therefore, extensive efforts have been made to understand the associated dynamics. The limited knowledge on ion trapping and detrapping largely impedes the development of superior electrochromic devices. In this manuscript, we stepped much deeper and succeeded to constructively visualize the ion trapping and detrapping dynamics in the model electrochromic material: amorphous WO₃ (Fig. R1, which is also Chart 1 of the main text). Meanwhile, we have also demonstrated that ion trapping and detrapping exist in all cathodic electrochromic oxides (Supplementary Fig.7 for Nb₂O₅ and Ta₂O₅, and papers of *Advanced Optical Materials* **2022**, 10, 2200903; *Materials Today Physics* **2023**, 30, 100958 (2023) for TiO₂). More importantly, we clarified the correlation between different spectral absorption and polaron hopping modes originated from different W valencies, thus provided a general picture of electrochromic mechanisms. Besides the experiment results, the methodologies used in this manuscript of a combination of spectroelectrochemistry and other spectro- and microscopic characterizations can be well expanded to other electrochromic materials and ion intercalated systems. Based on these points, we believe that this manuscript contains sufficient new results and insights to support its publication in *Nature Communications*.

Accordingly, we have re-ranged the wording in the **Abstract** (page 1) and **Introduction** (page 3) to make the advances of this manuscript more explicit.

		Species formation		Electrochromic feature
		2.0-4.0V	1.5-4.0V	
Reversibility	Reversible	Li _x WO ₃ (tungsten brnze) W ⁶⁺ ↔ W ⁵⁺ W ⁶⁺ ↔ W ⁴⁺		Full spectra absorption Short-wavelength absorption
	Releasable	shallow	Orthorhombic Li ₂ WO ₄	Non-absorbing, degrades colored state
		deep		Short-wavelength absorption, degrades both colored and bleached states
	Irreversible		W ⁴⁺ -Li ₂ WO ₄	Short-wavelength absorption

Fig. R1 General picture of the ion intercalation, trapping and detrapping in *a*-WO₃.

Comment 2: WO₃ is a very typical electrochromic materials and most thin film fabrication methods

can prepare a pretty high performance WO₃ thin film which can carry out electrochromic cycling for more than 10000 times, even 100000 times, without any degradation. The authors seem to deliberately use a higher voltage power 2.0-4.0 to degrade the WO₃. Only 5th or/and 20th cycles making the WO₃ degradation can not be equivalent to the real thousands cycles degradation. In addition, Li⁺ trapping problems is not very serious in WO₃ as like in NiO_x.

Response: It has been well recognized that ion trapping is one of the main reasons that limits the durability of the electrochromic materials, some representative papers are listed below:

Surf. Interface Anal. **1992**, *19*, 464-468;

J. Electrochem. Soc. **1991**, *138*, 2403-2408;

J. Electrochem. Soc. **1990**, *137*, 1300-1304;

Electrochim. Acta **2002**, *47*, 3977-3988;

Electrochim. Acta **2002**, *47*, 2435-2449

Phys. Rev. Lett. **2003**, *91*, 010602;

Nat. Mater. **2015**, *14*, 996-1001;

Adv. Mater. **2016**, *28*, 10518

Adv. Energy Mater. **2019**, *9*, 1902066

Appl. Surf. Sci. **2021**, *568*, 150898;

We agree that WO₃ is a very typical electrochromic material, therefore extensive efforts have been made to enhance the durability of WO₃ and some good performances have been achieved through annealing, nano-structure design, doping and electrolyte system selection, for example:

ACS Appl. Electron. Mater. **2023**, *5*, 5735-5748, which is also reference 25 in the revised version;

Solid State Sci. **2023**, *137*, 107127;

J. Phys.: Conf. Ser. **2023**, *2639*, 012028, which is also reference 27 in the revised version;

J. Solid State Electrochem. **2022**, *26*, 1667-1676;

Appl. Surf. Sci. **2022**, *582*, 152431;

ACS Appl. Mater. Interfaces **2021**, *13*, 11067-11077;

Sci. Rep. **2020**, *10*, 8430, which is also reference 26 in the revised version;

NPG Asia Mater. **2020**, *12*, 84;

In most of these works, obvious degradations can be seen after about 500 to 3000 cycles, respectively, even after various strategies have been used to enhance the durability. In papers *J. Phys.: Conf. Ser.* **2023**, *2639*, 012028 and *J. Solid State Electrochem.* **2022**, *26*, 1667-1676, the durability was reported to be largely-enhanced, for instance, the device stands for 100000 cycles and 15000 cycles, respectively. However, it can be found *i*) obvious degradation appeared, *ii*) the optical modulation was quite limited due to the small number of inserted Li ions. Moreover, it can also be found that the main electrochromic effect in the assembled device is from PB, rather than WO₃. For a single WO₃ thin film, charge density of ~35 mC cm⁻² was inserted and large optical modulation was achieved, however, no long-term durability was shown. The assembled device only

possessed charge density of $\sim 10 \text{ mC cm}^{-2}$ which is much less than the one inserted into single WO_3 film, thus the optical modulation is relatively smaller than single WO_3 film.

Overall, the intrinsic pain of durability of WO_3 was not cured to develop superior electrochromic devices with enhanced durability. In the main text, our deposition condition was chosen in order to study degradation and rejuvenation on a reasonable time scale (**Fig. 1c**). In fact, by controlling the sputtering parameters, it is able to obtain WO_3 films with different porosities whereas films with less porosity shows much slower degradation rate. As demonstrated in Supplementary **Fig. 6**, our film with less porosity showed that colored states degrade by only $\sim 14\%$ after 3000 cycles, and more importantly, it can also be rejuvenated. It is conclusive that, for WO_3 films using various enhanced strategies, the ion trapping needs to be well understood to develop superior electrochromic devices with enhanced durability.

The reviewer also expressed concerns on the potential window in our manuscript, *i.e.*, 2.0-4.0 V. We state that our choice is reasonable since cycling in this potential range under a sweep rate of 20 mV s^{-1} yields x of 0.65 in Li_xWO_3 . This is in excellent agreement with results in previous studies showing that Li^+ intercalation was reversible only when $x < 0.7$ for WO_3 (*J. Appl. Phys.* **2007**, *102*, 083538; *Nat. Mater.* **2015**, *14*, 996-1001). Under this cycling condition, the films have slow degradation of colored state and undegraded bleached state, and in accordance with common choice from others (*ACS Appl. Electron. Mater.* **2023**, *5*, 5735-5748; *Sci. Rep.* **2020**, *10*, 8430; *J. Phys.: Conf. Ser.* **2023**, 2639, 012028). As regards 1.5-4.0 V, it has been reported that serious degradation took place as the $x > 0.7$ (*J. Electrochem. Soc.* **1990**, *137*, 1300-1304; *J. Appl. Phys.* **2007**, *102*, 083538; *Nat. Mater.* **2015**, *14*, 996-1001). Therefore, we deliberately used a lower cut-off potential in order to *i*) find the potential limit for various traps and explore the possible match with other counter electrode over a larger potential window; *ii*) study the ion trapping and detrapping dynamics for deep and irreversible traps.

From our previous studies, Li trapping does not occur in NiO (anodic electrochromic oxide) since the electrochromism is a surface phenomenon in Li based electrolytes (*Adv. Funct. Mater.* **2015**, *25*, 3359-3370; *ChemElectroChem* **2016**, *3*, 266-275).

Overall, understanding the ion trapping and detrapping dynamics are essential to study the durability of WO_3 , and are representative for other electrochromic oxides. Therefore, this understanding holds significant importance in the development of superior electrochromic devices with enhanced durability.

Associated discussions of this part have been modified in **Introduction** (page 2, 3) and the discussion sections (page 5) in the revised version, to make the impacts of study on ion trapping and detrapping in WO_3 more highlighted and to well explain about the potential window.

Comment 3: One of the authors has published a series of papers on the trapping problems (Ref. 25, 30-32) around 2015. In this paper we do not see many new findings comparing with the old publications.

Response: As has been emphasized in our response to comment **1**, previous works were devoted to

phenomenal exploration, and the details of trapping and detrapping dynamics were not well understood, which largely hindered the development of superior electrochromic devices. In this manuscript, we stepped much deeper, and constructively visualized the ion trapping and detrapping dynamics in WO₃. Besides, we also demonstrated that ion trapping and detrapping exist in all cathodic electrochromic oxides (Supplementary Fig.7 for Nb₂O₅ and Ta₂O₅, and papers of *Advanced Optical Materials* **2022**, *10*, 2200903; *Materials Today Physics* **2023**, *30*, 100958 for TiO₂). Moreover, we clarified the correlation between different spectral absorption and polaron hopping modes originated from different W valencies. All these points have never been touched in the previous papers, thus contribute to the advances of this manuscript.

Thanks to the reviewer's comments, associated discussions of this part is now explicit in the *Abstract* (page 1) and the *Introduction* (page 3) in the revised version.

Comment 4: In figure 2 a, it shows a clear binding energy position shift of both W⁶⁺ and W⁵⁺, but there is no explanation of this shift found in the context.

Response: We sincerely appreciate the reviewer for pointing out our omission of the explanation of this binding energy position shift. The slight shift of certain valency upon ion insertion was also commonly observed by others (*Nat. Mater.* **2022**, *21*, 795-803; *Appl. Surf. Sci.* **2021**, *568*, 150898), which so far has not been well understood from the literature. However, considering the fact that this shift is commonly observed, we temporarily ascribe to coordination environment variation upon ion insertion and extraction.

We have added “One may note that it shows a nonnegligible binding energy position shift of both W⁶⁺ and W⁵⁺ (as well as W⁴⁺ in the next part in Fig. 3a) which was also observed from others^{37,53}. We temporarily ascribe this slight shift to coordination environment variation upon ion insertion and extraction” in p9.

Comment 5: The results from the measurement methods of XPS, Raman, TEM, spectral transmittance etc are not sufficient enough to support many of the statements in the paper such as the analysis and direct evidence of the three trapping modes.

Response: We appreciate the reviewer's comments; however, we respectfully disagree. Our characterizations by different techniques, for example, *in-situ* optical recording, XPS, Raman and TEM, are consistent and valid for all three trapping modes. We try to list the consistence and validness below:

Shallow traps originate from formation of orthorhombic Li₂WO₄ (JCPDS No: 28-0596) during long-term cycling, which is directly confirmed by TEM and XRD results. To prove the formation of orthorhombic Li₂WO₄, we investigated 10 more samples at their bleached states by high-resolution TEM and fast Fourier transform (FFT) patterns, all results are consistent (Supplementary Fig. 13), and these nano grains vanish after rejuvenation. Besides, XPS results confirmed that W remains +6 valency after ion trapping (Fig. 2a, 1000th bleached), which agrees well with our observation of the spectral transmittance at bleached states. Since W valency in Li₂WO₄ remains +6,

the results of TEM, XRD, XPS and spectral transmittance are all consistent. As revealed in **Fig. 2c** (pristine), in addition to a few broad peaks from amorphous WO_3 and ITO, Raman measurements shows strong background which leads to some weak vibration invisible. The absence of 282 cm^{-1} peak in Raman spectra of Li_2WO_4 was also reported by Takayanagi et al (*Appl. Surf. Sci.* **2021**, 568, 150898), which is analogous to our observation for shallow traps.

Deep traps are found to be multiple-step-determined, composed of mixed $\text{W}^{4+}\text{-Li}_2\text{WO}_4$, amorphous Li_2WO_4 and $\text{W}^{4+}\text{-Li}_2\text{O}$, which is evidenced by the consistent results of spectral transmittance, XPS, Raman and TEM. Specifically, formation of W^{4+} is undoubtedly confirmed by direct observation from the XPS (**Fig. 3a**) and Raman (**Fig. 3b**) results, agree well with the characteristic profile of spectral transmittance, namely, a centered absorption of short wavelength region raised from bipolaron hopping involved W^{4+} . Formation of Li_2WO_4 is also evidenced by clear peaks in Raman spectra emerged at $\sim 282\text{ cm}^{-1}$, and TEM results confirm that the Li_2WO_4 formed in deep traps is amorphous, this is in agreement with the very limited cycle numbers (20 cycles) in this case. The sharp peaks (comparing with the condition of shallow traps) indicate that the amorphous Li_2WO_4 are large in quantity which is due to two formation mechanisms: one is same as that in shallow traps, another is coupled with W^{4+} since the latter one can supply the extra oxygen for Li_2WO_4 through an O atom migration process (**Fig. 4a** of the main text, blue dash-line square). With a similar O atom migration process, $\text{W}^{4+}\text{-Li}_2\text{O}$ coupling is also formed (purple dash-line square in **Fig. 4a**), in which the formation of Li_2O is evidenced by XPS results.

Moreover, existence of all these species is impeccably consistent with and confirmed by the spectral transmittance, particularly among the detrapping process. Specifically, at the very beginning of the detrapping process (**Fig. 4b** of the main text), independent W^{4+} and $\text{W}^{4+}\text{-Li}_2\text{O}$ are released so the spectral transmittance slightly increases. This leads to decreased W^{4+} intensity and vanished Li_2O signals in XPS. The release of $\text{W}^{4+}\text{-Li}_2\text{O}$ is in agreement with its easy decomposition, as confirmed by the release of coupled $\text{W}^{4+}\text{-Li}_2\text{O}$ after potentiostatic detrapping at 4.0 V (Supplementary **Fig. 24**). As the detrapping process proceeds, a plateau of the spectral transmittance is observed because the release of amorphous Li_2WO_4 , the maintenance of W valencies in Li_2WO_4 and WO_3 causes the unvaried transmittance, this is also confirmed by the un-varied W 4f signal. The decreased Li 1s intensity in XPS, the recovered stretching modes of $\text{W}^{+6}\text{-O}$ and $\text{W}^{+6}=\text{O}$ in Raman spectra, as well as the notable rejuvenation (Supplementary **Fig. 22**) of the film after these plateau periods are all in good agreement. Finally, the spectral transmittance shows sharp increase and the current density profile upon potentiostatic detrapping shows an abrupt peak, indicate an intense release of trapped ions due to the decomposition of coupled $\text{W}^{4+}\text{-Li}_2\text{WO}_4$, as evidenced by the largely reduced W^{4+} intensity in XPS and Raman results. The increased intensity of Li_2WO_4 peak at this state is caused by the fluorescence vanishing due to the bleaching of the film after detrapping (*Appl. Surf. Sci.* **2021**, 568, 150898). The final release of a portion of W^{4+} accounted for the sharp increase of the optical transmittance. It should be noted these W^{4+} are coupled with Li_2WO_4

through O atom migration, thus making it difficult to be decomposed. The small amount of undecomposed W^{4+} - Li_2WO_4 stands after detrapping thus accounts for the irreversible traps (as verified by XPS, Raman and optical spectra at short wavelength). Synchronisms of their emergences, increases, reductions and residuals also confirm the coupling of W^{4+} - Li_2WO_4 . In conclusion, the results from *in-situ* optical recording, TEM, XPS and Raman are consistent, and our statements in the main text are well supported by these characterizations

Comment 6: In Figure 2a, the shown Li 1s dot signals can not see any signal peaks or any differences among all the lines except the 1000th colored state with a very small peak. Therefore the figure is non corresponding to the statements describing the changes of Li 1s signals.

Response: We sincerely thank for the reviewer's comments. We agree that the observed Li 1s signal is weak, which is caused by the low photoionization cross section of Li. In addition to direct observations of the intensity of Li 1s signals, it is more reasonable to discuss the amount of Li from the variation of W valencies (since the variation of W valencies is coupled with the amount of Li). Moreover, the signals of Li 1s, though weak, show really reasonable tendencies. For example, at the states of pristine, 5th bleached and after detrapping (**Fig. 2a**), there is no Li 1s signal observed; at the states of 5th colored, 1000th colored and 1000th bleached, Li 1s signals appeared and were reasonable in intensity tendencies. Specifically, at the states of 5th colored and 1000th bleached, Li 1s signals were obvious and originated from reversible sites and traps over the past 1,000 cycles, respectively. While at the state of 1000th colored, the Li 1s signal increases, because the contribution to the Li 1s signal here is both from inserted Li ions in the 1,000th cycle and accumulated Li ions in traps over the past 1,000 cycles. These reasonable tendencies indicate that the results are exact and valid.

Therefore, we calculated x value in Li_xWO_3 through the variation of W valencies, as we pointed out in **p9**. We have also emphasized the low detection limit of Li 1s in **p15** and **p19**.

REVIEWERS' COMMENTS

Reviewer #3 (Remarks to the Author):

The authors have supplied sufficient evidence and appropriate answers to referees' most concern and questions. I think it can be acceptable as the paper is now.